# The Utility of Narrow-Band Imaging International Colorectal Endoscopic Classification in Predicting the Histologies of Diminutive Colorectal Polyps Using I-Scan Optical Enhancement: A Prospective Study

**DOI:** 10.3390/diagnostics13162720

**Published:** 2023-08-21

**Authors:** Yeo Wool Kang, Jong Hoon Lee, Jong Yoon Lee

**Affiliations:** Division of Gastroenterology, Department of Internal Medicine, Dong-A University College of Medicine, Busan 49201, Republic of Korea; kangyw56@gmail.com (Y.W.K.); leejonghoon2002@gmail.com (J.H.L.)

**Keywords:** endoscopy, colonoscopy, colonic polyps, adenomatous polyps

## Abstract

(1) Background: This study aimed to evaluate the accuracy of predicting the histology of diminutive colonic polyps (DCPs) (≤5 mm) using i-scan optical enhancement (OE) based on the narrow-band imaging international colorectal endoscopic (NICE) classification. The study compared the diagnostic accuracy between experts who were already familiar with the NICE classification and trainees who were not, both before and after receiving brief training on the NICE classification. (2) Method: This prospective, single-center clinical trial was conducted at the Dong-A University Hospital from March 2020 to August 2020 and involved two groups of participants. The first group comprised two experienced endoscopists who were proficient in using i-scan OE and had received formal training in optical diagnosis and dye-less chromoendoscopy (DLC) techniques. The second group consisted of three endoscopists in the process of training in internal medicine at the Dong-A University Hospital. Each endoscopist examined the polyps and evaluated them using the NICE classification through i-scan OE. The results were not among the participants. Trained endoscopists were divided into pre- and post-training groups. (3) Results: During the study, a total of 259 DCPs were assessed using i-scan OE by the two expert endoscopists. They made real-time histological predictions according to the NICE classification criteria. For the trainee group, before training, the area under the receiver operating characteristic curves (AUROCs) for predicting histopathological results using i-scan OE were 0.791, 0.775, and 0.818. However, after receiving training, the AUROCs improved to 0.935, 0.949, and 0.963, which were not significantly different from the results achieved by the expert endoscopists. (4) Conclusions: This study highlights the potential of i-scan OE, along with the NICE classification, in predicting the histopathological results of DCPs during colonoscopy. In addition, this study suggests that even an endoscopist without experience in DLC can effectively use i-scan OE to improve diagnostic performance with only brief training.

## 1. Introduction

Colorectal cancer (CRC) is among the most commonly diagnosed cancers and a leading cause of cancer-related deaths [1]. Colonoscopy is recommended for detecting colonic polyps and has been established as the gold standard for CRC surveillance [2]. More than half of the polyps detected during surveillance colonoscopy are diminutive colorectal polyps (DCPs), that is, ≤5 mm in size [3,4]. However, white light endoscopy cannot accurately differentiate between adenomatous and hyperplastic DCPs, and it is currently standard practice to remove all polyps for subsequent histological analysis. Histopathological diagnosis of neoplastic polyps by endoscopic resection can lead to side effects, such as bleeding or perforation, and result in high medical costs [5,6]. Additionally, confirming the histological results for every polyp removed during the procedure is an expensive medical practice [7]. To address these issues, new endoscopic techniques have been proposed. One of these technologies is dye-less chromoendoscopy (DLC), and narrow-band imaging (NBI) is one of the most widely investigated DLC techniques [8,9,10,11].

NBI is an optical digital imaging technique wherein two selected wavelengths (415 + −30 nm and 540 + −30 nm) enhance the structural aspects of the surface of the mucosa and existing blood vessels [12]. Numerous studies have shown that NBI is capable of accurate tissue characterization [8,13,14,15]. This strong evidence led to the development of the NBI International Colorectal Endoscopic (NICE) classification, which allows for an accurate differentiation between adenomatous and hyperplastic polyps in vivo by analyzing the color, surface pattern, and vascular pattern of DCPs with NBI [16,17,18]. This classification employs NBI to classify colorectal polyps, accentuating vascular features. Utilizing NBI, the coloration, vessel arrangement, and surface presentation of polyps are evaluated to differentiate between hyperplastic polyps or sessile serrated lesions, adenomas or non-invasive carcinoma, and invasive carcinoma. Polyps are assigned to NICE types 1, 2, or 3, which correspond to hyperplastic polyps or sessile serrated lesions, adenomas or non-invasive carcinoma, and invasive carcinoma, respectively. With the advent of NBI, optical diagnosis is now possible, enabling the prediction of histopathological results without the need for confirmatory histology after resection, making strategies such as “resect and discard or leave behind” feasible [19,20].

In addition to NBI, blue laser imaging (BLI) and i-scan optical enhancement (OE) have demonstrated similar diagnostic rates [21,22,23]. The i-scan OE utilizes light that is restricted to specific bands to attain increased overall transmittance. This is accomplished by linking the highest points of the hemoglobin absorption spectrum (at 415, 540, and 570 nm), resulting in the creation of a seamless spectrum of wavelengths, which is almost identical to the principle of NBI [24]. Although i-scan OE operates on a similar principle as NBI, a few studies have suggested new diagnostic criteria, which are different from those of NBI [25,26,27,28]. However, they are not commonly used by many endoscopists. Additionally, the most widely used method among endoscopists for predicting the pathological outcome of DCPs still involves the use of the NICE classification. The purpose of this study was to evaluate the accuracy of predicting the histology of DCPs (≤5 mm) using i-scan OE based on the NICE classification. The study compared the diagnostic accuracy between experts who were already familiar with the NICE classification and trainees who were not, both before and after receiving brief training on the NICE classification.

## 2. Methods

### 2.1. Patients

This prospective, single-center clinical trial was conducted at the Dong-A University Hospital in Busan, Republic of Korea from March 2020 to August 2020. Patients aged between 20 and 75 years with DCPs detected during colonoscopy were included. Written informed consent was obtained from all patients prior to the procedure. Patients with poor bowel preparation, a history of colectomy, inflammatory bowel disease, or polyposis syndrome were excluded. Additionally, polyps with endoscopic signs of malignancy, as defined by the NICE classification, were not included in the study.

### 2.2. Study Participants

A total of five endoscopists participated and were divided into two distinct groups. The first group consisted of two experienced endoscopists who had received formal training in optical diagnosis and collectively performed over 2000 colonoscopies. Moreover, they had specialized knowledge in using i-scan OE for polyp characterization during colonoscopy and had extensive experience in performing DLC techniques. The term “expert group” was used to define these endoscopists.

On the other hand, the second group comprised three endoscopists who had received training in internal medicine at the Dong-A University Hospital and subsequently applied for a gastroenterology fellowship. These endoscopists had limited exposure to colonoscopy and had performed fewer than 50 procedures. Additionally, they had no prior experience or training in advanced endoscopic imaging methods, including DLC. The term “trainee group” was used to define these endoscopists.

### 2.3. Study Setting

Two expert endoscopists performed an endoscopy in each endoscopy room, and each examination room consisted of one expert endoscopist who directly performed the colonoscopy and three trainee endoscopists who were observed from behind. If a polyp was found during colonoscopy, it was recorded using high-definition white light (HDWL). Additionally, for this study, an image was recorded using the i-scan OE and categorized based on the NICE classification.

The recording of the NICE classification by trainee endoscopists was conducted in two steps. First, the trainees observed real-time examination images for a duration of two months without receiving any education on colorectal polyps. During this period, they recorded NICE classifications based on their observations. In the second step, the trainee endoscopists were educated on the principles of optical endoscopy, which included instructions on i-scan OE, basic knowledge of polyp classifications, and the NICE classification system. After completing the educational phase, the trainees continued to record the NICE classification using the same process as before. They observed real-time images in the examination room for another three months and documented the NICE classification based on their training. It is important to highlight that the NICE classification records of individual endoscopists were not shared in this study.

### 2.4. Training Module

The training module consisted of the following steps: First, three trainee endoscopists received a short 30 min training session explaining the NICE classification in optical diagnosis and how to differentiate between adenomas and hyperplastic polyps according to the NICE classification criteria. Next, they watched 30 representative static images of hyperplastic polyps and adenomas taken with the i-scan OE, while receiving additional training focused on interpreting the mucosal surface and blood vessel patterns. Following the training, the expert endoscopists provided direct feedback on the trainee endoscopists’ questions.

### 2.5. Colorectal Polyps

All colonoscopies were performed using a high-definition colonoscope equipped with a Pentax Medical OPTIVISTA EPK-I 7010 video processor (Pentax, Tokyo, Japan). For each polyp, a paired non-magnification, HDWL, and i-scan OE image were obtained. The i-scan OE was used to enhance the surface and vascular patterns, and the endoscopist made a real-time in vivo prediction of the underlying histological diagnosis based on color, surface pattern, and vessel pattern, as described for i-scan OE and using the NICE classification. The analysis indicated that DCPs could be accurately differentiated into hyperplastic and adenomatous polyps by i-scan OE in vivo, using color, surface pit pattern, and vascular pattern as differentiating criteria. When observed through i-scan OE, hyperplastic polyps appeared to be the same color or slightly lighter than the surrounding mucosa and lacked blood vessels or were presented as a single independent blood vessel. The polyp surface typically displayed a white or dot-like dark pattern. On the other hand, an adenomatous polyp is typically darker in color, appearing brown and more distinct from the surrounding mucosa. Blood vessels were distributed around the white structure in a brown pattern, and the surface of the polyp exhibited an oval, tubular, or curved pattern (Figure 1). After optical assessment, the polyps were resected using standard techniques and sent to an experienced gastrointestinal pathologist for histological assessment. Finally, the optical diagnosis was compared with histopathology as a reference standard.

### 2.6. End Points

The primary endpoint of this study was to determine the accuracy of predicting the histopathologic outcome of DCPs based on the NICE classification using i-scan OE. We compared the accuracy of experts who were already familiar with the NICE classification to that of trainees who were not. This comparison will be conducted separately for the periods before and after brief training on NICE classification. The secondary endpoint was to compare the sensitivity, specificity, negative predictive value (NPV), positive predictive value (PPV), and accuracy in each comparison.

### 2.7. Sample Size Estimation

Previous studies have shown that the accuracy of distinguishing adenomas from hyperplasia of DCPs through white-light endoscopy was 75–85%, while using NBI or i-scan OE increased the predictive value to approximately 90% [29,30,31,32,33,34]. This study set the pre-training accuracy of the trainee group using i-scan OE at 75% and the post-training accuracy of the trainee group using i-scan OE at 90%. We used the G*power program 3.1 (Faul, F., Erdfelder, E., Buchner, A., & Lang, A., 2009) to calculate the adequacy evaluation analysis for the number of statistical subjects, as well as the sample size calculation program based on Cohen’s sampling formula. The significance level α is 0.05, and the effect size is 0.25 (median). Calculated by a power of 0.80, a minimum of 100 polyps per group and a total of 200 polyps were needed.

### 2.8. Statistical Analysis

The primary aim of this study was to assess the diagnostic accuracy, including diagnostic sensitivity, specificity, positive predictive value (PPV), and negative predictive value (NPV), of i-scan OE for the in vivo prediction of polyp histology. Statistical analysis was performed using MedCalc Software for Windows version 17.1 (MedCalc Software Ltd., 2022, Oostende, Belgium). The diagnostic performance of i-scan OE was assessed by calculating the area under the receiver operating characteristic curve (AUROC) and 95% confidence interval (CI). The histopathology report was used as a reference for the validation of the endoscopic assessment.

## 3. Results

### 3.1. Study Population and Polyp Characteristics

A total of 120 patients were included, of which 81 were male and 39 were female participants. The number of patients included in the expert group and each pre- and post-training of the trainee group were relatively similar, and there was no significant difference in the ratio between male and female participants.

In total, 120 patients who exhibited 259 DCPs with an average size of 4.35 mm were included. Of the 259 polyps, 145 were located proximally and 114 were located distally. The clinical and demographic characteristics of the patients and the clinical and histopathological characteristics of the polyps are summarized in Table 1. The proportions of adenomas and hyperplastic polyps were 75.3% and 24.7%, respectively.

### 3.2. Pre- and Post-Training Results in Trainee Group Utilizing the NICE Classification through I-Scan OE

The pre-training test for the trainee group included 55 patients (38 males and 17 females). A total of 125 polyps were identified, with 68 polyps located proximally and 57 polyps located distally, accounting for 54.5% and 45.5%, respectively. The i-scan OE prediction results of the trainee group before training, according to the NICE classification criteria, were compared using histological examinations. The sensitivity values were 85.23%, 87.50%, and 85.23%, respectively, and the accuracy values were 81.60%, 81.60%, and 83.20%, respectively. The AUROC values in the trainee group before training were 0.791, 0.775, and 0.818, respectively.

The post-training test of the trainee group included 65 patients (43 males and 22 females). A total of 134 polyps were identified in these patients, with 77 polyps located proximally and 57 polyps located distally, accounting for 57.5% and 42.5%, respectively. When comparing the predicted results by the i-scan OE after training, based on the criteria of the NICE classification, to the histological examination, the sensitivity values were 98.13%, 97.19%, and 96.30%, respectively, and the accuracy values were 96.27% for all cases. The AUROC values in the trainee group after training were 0.935, 0.949, and 0.963, respectively (Table 2).

### 3.3. Comparison between Post-Training Trainee Group and Expert Group Results Utilizing the NICE Classification through I-Scan OE

A total of 259 polyps were assessed by i-scan OE in the steps of the study, in which two endoscopists expert in optical diagnosis made an in vivo differentiation between hyperplasia and adenomas based on the criteria of the NICE classification. When the expert endoscopists compared the results of adenomas and hyperplastic polyps predicted using i-scan OE based on histological examination results, the sensitivity and accuracy were 98.97% and 97.68%, respectively (Figure 2). Indeed, when the previously validated NBI criteria were applied to the first 259 polyps in this study, this allowed for the accurate discrimination of hyperplastic from adenomatous polyps. When comparing the AUROC values for predicting i-scan OE diagnosis between the expert and pre-training trainee group, the expert group achieved a value of 0.973, while the pre-training group achieved values of 0.791, 0.775, and 0.818, respectively. The *p*-value comparing the AUROC values was less than 0.05, indicating statistical significance. In addition, when comparing the expert and trainee groups after training, the AUROC value for predicting the diagnosis of i-scan OE increased to a level equivalent to that of the expert group, when compared to before training. No significant difference was observed in the statistical comparison between the AUROCs (Figure 3).

## 4. Discussion

The American Society of Gastrointestinal Endoscopy (ASGE) has introduced the Preservation and Incorporation of Valuable Endoscopic Innovations (PIVI) statement [35]. This statement establishes diagnostic thresholds that new endoscopic techniques must meet to ensure a reliable and accurate in vivo prediction of polyp histology. To determine whether DCPs can be discarded without histological evaluation, the ASGE suggests the following criteria: a surveillance interval agreement of over 90% based on histopathology and a negative predictive value exceeding 90%. In this study, the sensitivity, specificity, and overall diagnostic accuracies were 98.97%, 93.75%, and 97.68% in the expert group and 93.50%, 94.90%, and 96.27% in the post-training group, respectively, which were not statistically significant differences. This outcome surpassed the criteria set forth in the PIVI statement.

From these results, we identified several important implications of our study. First, the NICE classification, which is used to differentiate between adenomatous and hyperplastic polyps, was found to be applicable to i-scan OE. This suggests that the NICE classification developed for NBI can also be used for i-scan OE to predict the histology of DCPs. Second, it was observed that learning to differentiate between hyperplastic and adenomatous polyps using i-scan OE is easy. Moreover, this demonstrates a similar diagnostic efficacy between expert and trainee endoscopists. Even trainee endoscopists were able to effectively utilize i-scan OE after short 30 min training sessions. Taken together, this suggests that the i-scan OE displays a good diagnostic performance for the real-time histological prediction of DCPs and that a simple unified endoscopic classification, such as the NICE classification, has the potential for application in daily practice, regardless of the type of DLC technique.

DLC, which shares similar principles with NBI, includes techniques such as BLI and i-scan OE, which have been studied for the optical diagnosis of polyp tissues. BLI is an innovative technique that employs laser light to enhance endoscopic images. Unlike traditional methods that rely on xenon light, BLI harnesses the power of two distinct monochromatic lasers operating at 410 and 450 nm wavelengths [36]. Since the selective wavelength is similar to that of NBI, it was observed in a similar manner, and a comparative study of BLI showed a similar diagnostic accuracy to that of NBI [37]. A few studies have used i-scan OE to predict the histopathologic outcome of DCPs, but none have applied the NICE classification, which was first proposed and recognized by many endoscopists, and trained and applied it to inexperienced endoscopists. As evident in Figure 1, NBI and i-scan OE exhibit a high degree of similarity in their observations, as they share similar selected wavelengths to emphasize vascular structures, despite minor differences in color tones. In fact, predictions of pathological outcomes through i-scan OE for DCPs surpassed 90%, closely resembling the results obtained through previous NBI-based predictions. These results suggest that i-scan OE is suitable for a reliable endoscopic characterization and differentiation of hyperplasia and adenomatous polyps.

However, this study had a few limitations. Although we measured the sample size to satisfy the hypotheses of the study, two experts and three trainees from a single institution participated in this study. A study with a larger sample size and involving more endoscopists is needed. It is important to note that while i-scan OE shows promising potential for characterizing colorectal polyps, its direct clinical applicability has not yet been evaluated in larger studies. This study highlights the need for further research to determine whether optical diagnosis using i-scan OE leads to actual cost savings and whether it can effectively narrow the colonoscopy gap [38,39]. Patient acceptance of optical diagnosis is currently limited, but if research results demonstrate an excellent predictive value, a follow-up period based on optical endoscopy results can be established.

Nevertheless, this study is of great significance, as it is the first one to compare the pre- and post-training of trainees by applying an already widely used diagnostic method called the NICE classification to i-scan OE, a principle similar to NBI. Based on these findings, the application of the NICE classification for predicting the histology of DCPs using i-scan OE was shown to be diagnostically accurate. Moreover, this study demonstrated that the diagnostic value achieved by learning the NICE classification using i-scan OE was comparable to that of the expert group with only a short training program.

## 5. Conclusions

In summary, this study highlights the potential of i-scan OE, along with the NICE classification, in predicting the histology of DCPs during colonoscopy. This shows that experienced endoscopists proficient in i-scan OE can achieve a high accuracy in differentiating between adenomatous and hyperplastic polyps. Furthermore, the study suggests that even inexperienced endoscopists can be quickly trained to effectively utilize i-scan OE, leading to an improved diagnostic performance.

## Figures and Tables

**Figure 1 diagnostics-13-02720-f001:**
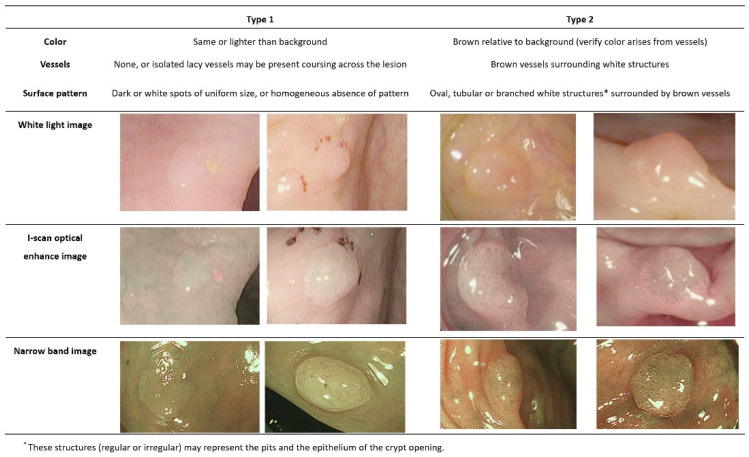
The polyps observed through narrow-band image and i-scan optical enhancement are classified according to narrow-band image international colorectal endoscopic classification.

**Figure 2 diagnostics-13-02720-f002:**
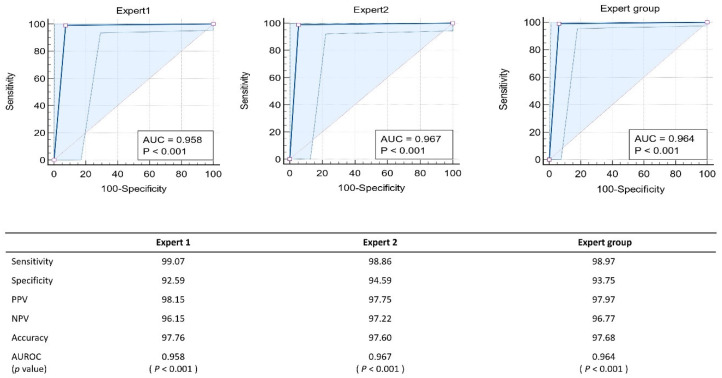
AUROC evaluates the diagnostic predictive value of diminutive colorectal polyps in expert endoscopists utilizing the narrow-band imaging international colorectal endoscopy classification using i-scan optical enhancement.

**Figure 3 diagnostics-13-02720-f003:**
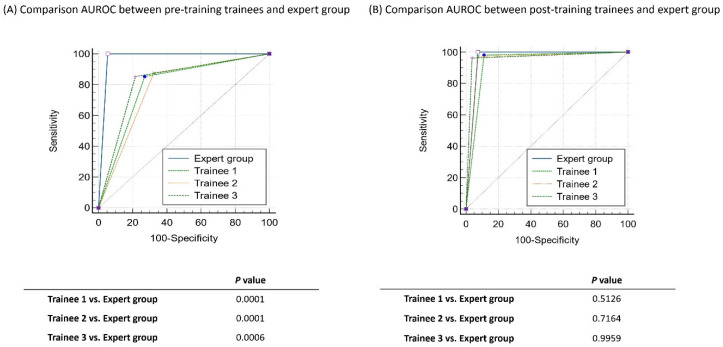
Comparison of AUROC between pre- and post-training trainees and expert group utilizing the narrow band imaging international colorectal endoscopy classification using i-scan optical enhancement.

**Table 1 diagnostics-13-02720-t001:** Clinical and demographic characteristics of the patients and polyps.

Patients Characteristics	All	Trainee Endoscopists Group	
		Pre-Training	Post-Training
Total, n	120	55	65
Sex, n (%)			
Male	81 (67.5)	38 (69.1)	43 (66.67)
Female	39 (32.5)	17 (30.9)	22 (33.33)
**Polyp Characteristics**			
Total, n	259	125	134
Location, n (%)			
Proximal	145 (56.0)	68 (54.5)	77 (57.5)
Distal	114 (44.0)	57 (45.5)	57 (42.5)
Histology, n (%)			
Adenoma	195 (75.3)	88 (70.4)	107 (79.9)
Hyperplastic	64 (24.7)	37 (29.6)	27 (20.1)

**Table 2 diagnostics-13-02720-t002:** Pre- and post-training results in trainee group utilizing the NICE classification through i-scan optical enhancement.

		Sensitivity % (95% CI)	Specificity % (95% CI)	PPV % (95% CI)	NPV % (95% CI)
Pre-training	**1**	85.98 (77.9–91.9)	81.48 (61.9–93.7)	94.8 (89.3–97.6)	59.5 (47.0–70.8)
	**2**	86.92 (79.0–92.7)	74.07 (53.7–88.9)	93.0 (87.5–96.2)	58.8 (45.5–71.0)
	**3**	87.85 (80.1–93.4)	85.19 (66.3–95.8)	95.9 (90.5–98.3)	63.9 (50.9–75.1)
Post-training	**1**	97.73 (92.0–99.7)	91.89 (78.1–98.3)	96.6 (90.6–98.8)	94.4 (81.1–98.5)
	**2**	98.86 (93.8–100.0)	91.89 (78.1–98.3)	96.7 (90.7–98.8)	97.1 (82.9–99.6)
	**3**	100.00 (95.9–100.0)	94.59 (81.8–99.3)	97.8 (92.0–99.4)	100.0

NICE—the narrow-band imaging international colorectal endoscopic; CI—confidence interval; PPV—positive predictive value; NPV—negative predictive value; OE—optical enhancement.

## Data Availability

The data presented in this study are available on request from the corresponding author. The data are not publicly available due to restrictions aimed at protecting patient confidentiality.

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
