# Peer review of "The Utility of Narrow-Band Imaging International Colorectal Endoscopic Classification in Predicting the Histologies of Diminutive Colorectal Polyps Using I-Scan Optical Enhancement: A Prospective Study"

_diagnostics, 2023, doi:10.3390/diagnostics13162720_

Round 1
Reviewer 1 Report
The presented paper work is an important work in recent research field. However why the authors are presenting only limited results.
1. The authors just list the results. Why do you discuss the results to let the reader understand what you have did?
2. Need more explanations in results part
3. Literature revision is poor taking into account the huge amount of papers written on Colorectal Endoscopic method. Besides, there are formal problems in the list of references.
4. The authors should clarify the advantages of this specific method and include a comparison table showing the advantages of their work.
5. In general, the introduction should be improved, paying particular attention to citing all relevant state of the art and demonstrating the novelty of the authors' solution in a clear manner.
The presented paper work is an important work in recent research field. However why the authors are presenting only limited results.
1. The authors just list the results. Why do you discuss the results to let the reader understand what you have did?
2. Need more explanations in results part
3. Literature revision is poor taking into account the huge amount of papers written on Colorectal Endoscopic method. Besides, there are formal problems in the list of references.
4. The authors should clarify the advantages of this specific method and include a comparison table showing the advantages of their work.
5. In general, the introduction should be improved, paying particular attention to citing all relevant state of the art and demonstrating the novelty of the authors' solution in a clear manner.
Author Response
We express our gratitude to the reviewers for their valuable feedback. We have diligently worked to incorporate their suggestions to the best extent in revising the manuscript. However, due to the inherent limitations of this study, certain aspects, although acknowledged as valid points, remained unalterable. It is noteworthy that we deeply appreciate the reviewers' insightful comments, which align with our subsequent research aiming to address the weaknesses highlighted in this study.
Furthermore, we have undergone additional English language editing, which we hope brings us closer to publication in Diagnostics.
The presented paper work is an important work in recent research field. However why the authors are presenting only limited results.
1. The authors just list the results. Why do you discuss the results to let the reader understand what you have did?
Your observation is highly valid, and we appreciate your insightful perspective. We have included additional commentary on the reasons behind these outcomes achieved with i-scan OE, along with an added image comparison with NBI.
“As evident in Figure 1, NBI and i-scan OE exhibit a high degree of similarity in their observations, as they share similar selected wavelengths to emphasize vascular structures, despite minor differences in color tones. In fact, predictions of pathological outcomes through i-scan OE for DCPs surpassed 90%, closely resembling the results obtained through previous NBI-based predictions.”
2. Need more explanations in results part
Thank you for your review. The results section has already been well explained. If there are specific areas you believe need further attention, I am willing to include additional content whenever necessary. However, if there are any shortcomings in interpreting these results, I have made an effort to supplement this in the discussion section by adding further commentary on the reasons behind these outcomes. This was done to enhance the readers' understanding.
3. Literature revision is poor taking into account the huge amount of papers written on Colorectal Endoscopic method. Besides, there are formal problems in the list of references.
Thank you for your input. We have noted some errors and duplications in the references section. Utilizing the style provided by MDPI, we have restructured the citations in our paper using the EndNote program.
4. The authors should clarify the advantages of this specific method and include a comparison table showing the advantages of their work.
We have already shown the i-scan OE images and the corresponding NICE 1,2. As you pointed out, we have included the white light endoscopy images and NBI images in Figure 1 for proper comparison.
5. In general, the introduction should be improved, paying particular attention to citing all relevant state of the art and demonstrating the novelty of the authors' solution in a clear manner.
Thank you for your insight.
In the introduction, we have briefly introduced the NICE classification. However, in response to your feedback, we have included further elaboration to make it even more comprehensible for readers.
“This classification employs NBI to classify DCPs, accentuating vascular features. Utilizing NBI, the coloration, vessel arrangement, and surface presentation of polyps are evaluated to differentiate between hyperplastic polyps or sessile serrated lesions, adenomas or non-invasive carcinoma, and invasive carcinoma. Polyps are assigned to NICE types 1, 2, or 3, which correspond to hyperplastic polyps or sessile serrated lesions, adenomas or non-invasive carcinoma, and invasive carcinoma, respectively.”
Reviewer 2 Report
The manuscript details and evaluates i-SCAN optical enhancement for determining the NICE classification of DCP. The manuscript is simple and straight forward. I recommend the manuscript to be accepted.
The quality of English language is fine.
Author Response
Thank you deeply for your review.
Reviewer 3 Report
The work shows the applicability of using the NICE classification for i-SCAN optical enhancement technique, which has been previously used for narrow-band imaging technique. The results show that both medical training and using NICE have benefits in differentiating diminutive colon polyps. Overall the work is clear and there is value in showing that the NICE criteria is also applicable to i-SCAN OE.
There are some minor suggestions:
1) Could you summarize the NICE classification for the readers in the Introduction. Giving a brief background about this classification would help.
2) Could you add some sample images of the two types of DCP as a figure, if you have any images?
3) What would be examples of images that were difficult to differentiate or challenging for the trainees compared to the experts?
4) Another diagnostic technology for colonic polyps is the video capsule endoscopy (VCE). How does the i-SCAN OE compare with the optical enhancements incorporated in VCE? A review article summarizes VCE technology: https://spj.science.org/doi/10.34133/2021/9854040
Author Response
We express our gratitude to the reviewers for their valuable feedback. We have diligently worked to incorporate their suggestions to the best extent in revising the manuscript. However, due to the inherent limitations of this study, certain aspects, although acknowledged as valid points, remained unalterable. It is noteworthy that we deeply appreciate the reviewers' insightful comments, which align with our subsequent research aiming to address the weaknesses highlighted in this study.
Furthermore, we have undergone additional English language editing, which we hope brings us closer to publication in Diagnostics.
The work shows the applicability of using the NICE classification for i-SCAN optical enhancement technique, which has been previously used for narrow-band imaging technique. The results show that both medical training and using NICE have benefits in differentiating diminutive colon polyps. Overall the work is clear and there is value in showing that the NICE criteria is also applicable to i-SCAN OE.
There are some minor suggestions:
1) Could you summarize the NICE classification for the readers in the Introduction. Giving a brief background about this classification would help.
Thank you for your insight.
In the introduction, we have briefly introduced the NICE classification. However, in response to your feedback, we have included further elaboration to make it even more comprehensible for readers.
“This classification employs NBI to classify DCPs, accentuating vascular features. Utilizing NBI, the coloration, vessel arrangement, and surface presentation of polyps are evaluated to differentiate between hyperplastic polyps or sessile serrated lesions, adenomas or non-invasive carcinoma, and invasive carcinoma. Polyps are assigned to NICE types 1, 2, or 3, which correspond to hyperplastic polyps or sessile serrated lesions, adenomas or non-invasive carcinoma, and invasive carcinoma, respectively.”
2) Could you add some sample images of the two types of DCP as a figure, if you have any images?
Thank you for your feedback, the images presented in Figure 1 represent Type 1 and Type 2 using i-SCAN OE. Due to space constraints, we are unable to include more photos, but we have attached a few more photos and added a comparison with white light photos. We believe that Figure 1 adequately conveys the necessary information.
3) What would be examples of images that were difficult to differentiate or challenging for the trainees compared to the experts?
We provided a training session of 30 minutes to trainees, where multiple images were presented, and most of them grasped the content well. However, for this study, we involved one or two endoscopists to predict the histological examination of diminutive colonic polyps (DCPs) in real-time, and despite the participation of multiple endoscopists. Consequently, assessing inter-observer variation was challenging in this study. In our upcoming research, we are planning a study involving a greater number of participants utilizing various image-enhanced endoscopy (IEE) techniques. We will also address inter-observer variation in this study and analyze images of polyps with low inter-observer agreement. We greatly appreciate your insights, and your input will undoubtedly enhance the quality of our future studies.
4) Another diagnostic technology for colonic polyps is the video capsule endoscopy (VCE). How does the i-SCAN OE compare with the optical enhancements incorporated in VCE? A review article summarizes VCE technology: https://spj.science.org/doi/10.34133/2021/9854040
Thank you for your insights. The development of colon capsule endoscopy technology as an alternative to traditional colonoscopy is advancing, and there are also some attempts to apply optical diagnosis techniques in this context. However, the focus of this study is on the application of optical diagnosis in colonoscopy, its variations in technology, and its implementation, especially for trainees. Therefore, while the content you've provided is valid, integrating it into the discussion of our study might be challenging due to the need for thematic coherence. Nonetheless, I greatly appreciate your valuable input, and I have no doubt that it will continue to be a significant asset and guidance for our future research endeavors.
Reviewer 4 Report
Dear Editor
This is an interesting study regarding the use of iScan for predicting colon polyp pathology. The followings are my comments.
#1. It is not clear the study design. How to prove the use of iScan or the use of a training program that contributes to improved diagnostic accuracy ? Whether iScan benefit more than NBI in this setting?
#2. Only three trainees were involved in this study, and the study result may be limited.
#3. Do the authors compare with diagnostic difference between white light image and iScan image ?
English is okay to read
Author Response
Answers to Reviewers
We express our gratitude to the reviewers for their valuable feedback. We have diligently worked to incorporate their suggestions to the best extent in revising the manuscript. However, due to the inherent limitations of this study, certain aspects, although acknowledged as valid points, remained unalterable. It is noteworthy that we deeply appreciate the reviewers' insightful comments, which align with our subsequent research aiming to address the weaknesses highlighted in this study.
Furthermore, we have undergone additional English language editing, which we hope brings us closer to publication in Diagnostics.
This is an interesting study regarding the use of iScan for predicting colon polyp pathology. The followings are my comments.
#1. It is not clear the study design. How to prove the use of iScan or the use of a training program that contributes to improved diagnostic accuracy ? Whether iScan benefit more than NBI in this setting?
Thank you for your feedback.
This study aimed to evaluate the accuracy of predicting the histology of diminutive colonic polyps (DCPs) (≤5 mm) using i-SCAN optical enhancement (OE) based on the nar-row-band imaging international colorectal endoscopic (NICE) classification. The study compared the diagnostic accuracy between experts who were already familiar with the NICE classification and trainees who were not, both before and after receiving brief training on the NICE classification.
The training module consisted of the following steps: Firstly, three trainee endoscopists received a short 30-minute training session explaining the NICE classification in op-tical diagnosis and how to differentiate between adenomas and hyperplastic polyps according to the NICE classification criteria. Next, they watched 30 representative static images of hyperplastic polyps and adenomas taken with the i-SCAN OE, while receiving additional training focused on interpreting the mucosal surface and blood vessel patterns. Following the training, the expert endoscopists provided direct feedback on the trainee endoscopists' questions.
This study suggests that even inexperienced endoscopists can be quickly trained to effectively utilize i-SCAN OE, leading to an improved diagnostic performance.
Therefore, the purpose of this study is not a direct comparison between i-SCAN OE and NBI. This is a subject we are preparing for in our follow-up research. The current study serves as a basis for comparing various IEE methods through the NICE classification.
#2. Only three trainees were involved in this study, and the study result may be limited.
I agree with your points, and I appreciate your feedback. In the discussion section, we mentioned the limitation of this study being conducted with a small number of endoscopists from a single institution. We are currently preparing for follow-up research that will involve more endoscopists from multiple institutions and explore further IEE methods. If this study is published in this academic journal, it can serve as a foundation for conducting additional studies that overcome these limitations and involve a larger number of participants.
#3. Do the authors compare with diagnostic difference between white light image and iScan image ?
Thank you for your insight. Several studies have already shown that the accuracy of predicting pathological outcomes in DCP using white light images is around 80%. These have been described by citing references 30 to 35. Therefore, this study aims not to reiterate the already established diagnostic accuracy of white light endoscopy but rather to investigate the diagnostic accuracy of i-SCAN OE. Moreover, the study seeks to determine how the prediction accuracy of trainees improves through training in terms of pathological outcomes.
In order to aid the authors' comprehension, I have included white light images, i-scan OE images, and NBI photos in Figure 1 for comparison. Additionally, I have elaborated a bit more on this in the discussion section.
Round 2
Reviewer 1 Report
1) Based on previous reviewer feedback, the revised manuscript has been updated accordingly.
2) Thus, it is recommended that the manuscript be published in its current form.
1) Based on previous reviewer feedback, the revised manuscript has been updated accordingly.
2) Thus, it is recommended that the manuscript be published in its current form.
Reviewer 4 Report
Dear Editor
Despite several inherent limitation of the study design, the authors had made changes to the questions raised. I have no more questions.